# PARTNER-AWARE HIERARCHICAL SKILL DISCOVERY FOR ROBUST HUMAN-AI COLLABORATION

## ABSTRACT

Multi-agent collaboration, especially in human-AI (HAI) teaming, requires agents that can adapt to novel partners with diverse and dynamic behaviors. Conventional Deep Hierarchical Reinforcement Learning (DHRL) methods focus on agent-centric rewards and overlook partner behavior, leading to shortcut learning, where skills exploit spurious information instead of adapting to partners' dynamic behaviors. This limitation undermines agents' ability to adapt and coordinate effectively with novel partners. We introduce Partner-Aware Skill Discovery (PASD), a DHRL framework learning skills conditioned on partner behavior. PASD introduces a contrastive intrinsic reward to capture patterns emerging from partner interactions, aligning skill representations across similar partners while maintaining discriminability across diverse strategies. By structuring the skill space based on partner interactions, this approach mitigates shortcut learning and promotes behavioral consistency, enabling robust and adaptive coordination. We conduct extensive evaluations in Overcooked-AI across three complementary settings: (1) a diverse self-play partner population spanning a wide range of skill levels and play styles, (2) human proxy partners trained from real human–human trajectories, and (3) a controlled human-subject study with 25 participants. PASD consistently outperforms existing population-based and hierarchical baselines, demonstrating transferable skill learning that generalizes across a wide range of partner behaviors. Analysis of learned skill representations shows that PASD adapts effectively to diverse partner behaviors, highlighting its robustness in HAI collaboration.

## 1 INTRODUCTION

Developing intelligent agents that can coordinate effectively with humans and other novel partners has long been a central challenge in multi-agent reinforcement learning (MARL) (Klein et al., 2004; Alami et al., 2006; Bard et al., 2020). Unlike adversarial settings (Ye et al., 2020), where success is measured by outperforming an opponent, collaboration is far more challenging as it requires adapting to novel partners with diverse and often unpredictable behaviors (Hu et al., 2020). Early approaches relied on behavior cloning (BC) from human–human interaction data, but these methods are costly, time-consuming, and struggle to capture the diversity of real-world behaviors (Carroll et al., 2019). Furthermore, even in fully observable environments, partner behaviors are not fully predictable from a single state. Differences in timing, hesitation, movement rhythms, and style preferences unfold over sequences of actions, requiring temporal modeling to capture and adapt to these patterns. Conditioning on the current state is insufficient to avoid behavioral interference or forgetting of previously learned coordination strategies. More recently, hierarchical reinforcement learning (HRL) has advanced collaboration by decomposing complex tasks into reusable skills, enabling more structured exploration and improved coordination (Eysenbach et al., 2019; Loo et al., 2023). HRL provides a framework for structuring agent behavior through temporally extended actions, or 'skills', which can capture reusable patterns of behavior. By learning a set of diverse and reusable skills, agents can explore more efficiently and adapt their behavior in complex environments.

However, standard skill discovery methods remain largely agent-centric, optimizing for diversity without accounting for partner influence. Consequently, the learned skills may support individual performance but are insufficient for robust coordination with diverse partners. We argue that this limitation arises from the agent-centric nature of reward optimization. Existing approaches maximise expected returns from the agent's perspective, often ignoring the influence of the partner on

cooperative dynamics. This misalignment leads to *shortcut learning* (Wei et al., 2023), where agents exploit spurious correlations in the environment rather than capturing information relevant for partner interactions. As a result, agents develop behaviors that prioritize their own action diversity but fail to generalize coordination across novel partners.

Prior skill discovery methods in HRL often maximize mutual information (MI) between skills and observational states as a proxy for behavioral diversity Gregor et al. (2016); Eysenbach et al. (2019). While this encourages the agent to develop distinguishable behaviors, the objective is bounded by the entropy of skills and does not ensure sensitivity to partner behavior. As a result, these approaches often learn simple and static skills with limited adaptability, leading to poor state coverage and weak coordination, as highlighted in recent studies (Campos et al., 2020; Jiang et al., 2022). Moreover, tractable variational estimators of MI (Eysenbach et al., 2019), typically implemented with neural networks optimized via cross-entropy or related objectives, are prone to shortcut learning . In practice, they often capture spurious correlations in state features rather than the interaction patterns relevant for effective collaboration Wei et al. (2023).

We introduce Partner-Aware Skill Discovery (PASD), an HRL approach for learning skills that adapt to diverse collaborator behaviors. PASD maximizes a variational lower bound on MI between skills and sub-trajectories, encouraging representations that are consistent and reproducible across partner interactions. This is achieved via a contrastive objective that ensures skill representations are discriminative across heterogeneous partners while remaining consistent for partners with similar behaviors. By capturing patterns shaped by partner behavior rather than agent-centric correlations, PASD mitigates shortcut learning and produces skills that generalize across partners, supporting effective partner-adaptive coordination.

In summary, our work makes the following key contributions. First, we introduce PASD, a DHRL framework that enables robust human-AI coordination through skill representations conditioned on partner behavior. Second, a novel contrastive intrinsic reward is proposed and incorporated into PASD to encourages consistency in skill representations across similar partners by capturing shared patterns from parallel rollouts while maintaining discriminability across diverse partner behaviors. This intrinsic reward, derived from contrastive learning, encourages behavioral diversity across diverse partners, mitigating shortcut learning caused by spurious state information, and directly leveraging partner-relevant information from the observational space. We further emphasize that PASD leverages HRL to mitigate forgetting by learning separate latent skills and captures temporal patterns of partner behavior, enabling robust adaptation to diverse human-like coordination styles even in fully observable environments. Finally, we extensively evaluate PASD in the Overcooked-AI environment, partnering the agent with a diverse self-play population spanning multiple skill levels and play styles, as well as with human-proxy models trained from human–human demonstrations. In addition, a controlled human-subject study with real human participants further shows that PASD yields significantly higher joint rewards than existing approaches, demonstrating that PASD learns transferable skills that generalize effectively across a wide range of partners, enabling robust and adaptive human-AI coordination.

## 2 RELATED WORK

Recent work has explored building agents that can coordinate with human partners Carroll et al. (2019); Hao et al. (2024). Carroll et al. Carroll et al. (2019) introduced the *Overcooked-AI* environment and trained PPO agents with human proxy models from human gameplay. While improving robustness, this requires extensive and costly human data. Hao et al. Hao et al. (2024) introduce intrinsic rewards to encourage agents to explore states that yield sparse rewards when coordinating with human proxy models. Strouse et al. Strouse et al. (2021) propose Fictitious Co-Play (FCP), generating a pool of self-play policies and past versions to train adaptive agents without human data. Some works further improve partner heterogeneity using entropy-based objectives during training (Lupu et al., 2021; Garnelo et al., 2021; Zhao et al., 2023; Loo et al., 2023). Hidden-utility Self-Play HSP (Yu et al., 2023) extends FCP by modeling human biases as hidden reward functions, generating a diverse policies to train adaptive agents that can cooperate with unseen humans with preferences deviating from environment rewards. Jha et al. Jha et al. (2025) propose Cross-Environment Cooperation (CEC), which trains agents across a distribution of environments to acquire general cooperative skills, enabling zero-shot coordination with novel partners. While effective for generalization across

tasks, CEC does not explicitly model partner-adaptive skill discovery within a single environment, which is the focus of our method. While these methods focus on learning a single-level agent policy, effective human-AI coordination requires reasoning over temporally extended behaviors and adapting to partners with diverse and evolving strategies.

HRL provides a framework for reasoning over temporally extended behaviors, making it well-suited for multi-agent and human-AI coordination. By learning policies at multiple temporal levels (Sutton et al., 1999; Flet-Berliac, 2019; Pateria et al., 2021), HRL captures high-level strategic planning and low-level execution. Classical approaches such as options (Bacon et al., 2017; Eysenbach et al., 2019) and feudal learning (Vezhnevets et al., 2017) illustrate temporal hierarchy benefits, extended to cooperative multi-agent settings in recent work (Loo et al., 2023; Yang et al., 2023a). Methods like DIAYN (Eysenbach et al., 2019) encourage diverse behaviors using intrinsic rewards maximizing MI between skills and states/actions. However, these agent-centric approaches are prone to shortcut learning, capturing spurious patterns instead of partner-relevant behaviors. Hierarchical Population Training (HIPT) (Loo et al., 2023) adapts HRL to human-AI coordination by shaping the high-level policy via influence-based intrinsic rewards but trains the low-level policy only on extrinsic rewards, risking skill collapse. Our approach introduces a novel intrinsic reward to mitigate shortcut learning, ensuring behavioral consistency across similar partners while remaining discriminative to diverse strategies, supporting adaptive human-AI coordination

## 3 PRELIMINARIES

We consider a multi-agent setting in which two agents collaborate to complete shared tasks, with the framework naturally extending to settings involving more agents. One agent is controlled by a learning policy $\pi_\theta(\cdot \mid s_t)$, while the other is governed by a partner policy $\pi^p(\cdot \mid s_t)$, sampled uniformly from a population of pretrained partners $\mathcal{D}_p$ at the start of each episode. The objective is to train $\pi_\theta(\cdot \mid s_t)$ to achieve high returns when paired with novel partners drawn from a separate evaluation distribution $\mathcal{D}'_p$. The environment is modeled as a two-player Markov game $\mathcal{M} = (\mathcal{S}, \mathcal{A}, \mathcal{A}^p, \mathcal{P}, r, \gamma, \rho_0)$, where $\mathcal{S}$ is the state space, $\mathcal{A}$ and $\mathcal{A}^p$ are the action spaces of the learning agent and the partner, $\mathcal{P}$ is the transition kernel, $r : \mathcal{S} \times \mathcal{A} \times \mathcal{A}^p \to \mathbb{R}$ is the shared team reward, $\gamma \in (0, 1)$ is the discount factor, and $\rho_0$ is the initial state distribution. At each timestep $t$, the learning agent selects an action $a_t \sim \pi_\theta(\cdot \mid s_t)$, while the partner executes $a_t^p \sim \pi^p(\cdot \mid s_t)$, and the next state is drawn from $s_{t+1} \sim \mathcal{P}(s_{t+1} \mid s_t, a_t, a_t^p)$.

From the perspective of the learning agent, the effective dynamics marginalize over both the partner's stochasticity and the population distribution:

$$\mathcal{P}_{\mathcal{D}_p}(s' \mid s, a) = \mathbb{E}_{\pi^p \sim \mathcal{D}_p} \mathbb{E}_{a^p \sim \pi^p(\cdot \mid s)} \big[ \mathcal{P}(s' \mid s, a, a^p) \big].$$

This formulation emphasizes that the agent must learn a policy that is robust to variations in partner behavior while maximizing expected returns over the population of collaborators. For a given partner $\pi^p$, the return is

$$J(\pi_\theta \mid \pi^p) = \mathbb{E}\left[ \sum_{t=0}^{\infty} \gamma^t r(s_t, a_t, a_t^p) \right]. \tag{1}$$

To encourage robustness to novel partners, the learning objective is the expected return over the partner population:

$$J(\pi_\theta) = \mathbb{E}_{\pi^p \sim \mathcal{D}_p} \big[ J(\pi_\theta \mid \pi^p) \big]. \tag{2}$$

**Hierarchical reinforcement learning:** In collaborative multi-agent environments, effective coordination requires reasoning over temporally extended behaviors and adapting to partners with diverse and dynamically changing strategies. To capture these aspects, we model the learning agent using a hierarchical policy inspired by the options framework Sutton et al. (1999). Formally, we consider DHRL setup with a high-level manager $\pi_{hi}(z \mid s)$ and a low-level controller $\pi_{lo}(a \mid s, z)$. The high-level manager selects latent skills $z \in \mathcal{Z}$, which guide temporally extended behaviors executed by the low-level controller. Each skill $z_k$ is executed over a segment from $t_k$ to $t_{k+1} - 1$, producing cumulative segment reward:

$$\mathcal{R}^Z(s_{t_k}, z_k) = \sum_{t=t_k}^{t_{k+1}-1} \gamma^{t-t_k} r(s_t, a_t, a_t^p), \quad z_k \sim \pi_{hi}(\cdot \mid s_{t_k}), \ a_t \sim \pi_{lo}(\cdot \mid s_t, z_k), \tag{3}$$

where $a_t^p$ denotes the partner's action at time $t$ and $\gamma \in [0, 1]$ is a discount factor. The high-level objective is the expected return over all skill segments:

$$J_{hi}(\pi_{hi}, \pi_{lo} \mid \pi^p) = \mathbb{E}\left[ \sum_{k=0}^{K-1} \gamma^{t_k} \mathcal{R}^Z(s_{t_k}, z_k) \right]. \tag{4}$$

Skill execution is controlled by a stochastic termination function $\beta(z, s)$, which determines whether the current skill continues or a new skill should be selected. This allows the manager to adaptively update skills at irregular intervals $T_{hi}$ based on the evolving collaborative context.

Within each skill segment, the low-level controller $\pi_{lo}(a \mid s, z)$ outputs primitive actions conditioned on the current state and the active skill. The low-level policy is trained to reliably realize the intended skill, producing sequences of actions that induce state transitions $s \to s'$ through the environment dynamics. Formally, the low-level objective can be expressed as:

$$J_{lo}(\pi_{lo} \mid z, \pi^p) = \mathbb{E}\left[ \sum_{t=t_k}^{t_{k+1}-1} \gamma^{t-t_k} r(s_t, a_t, a_t^p) \right], \tag{5}$$

where the expectation is conditioned on the currently active skill $z_k$. The high-level manager, low-level controller, and termination function together define the joint hierarchical policy $\pi = (\pi_{hi}, \pi_{lo}, \beta)$, which is optimized end-to-end via proximal policy optimization (PPO) algorithm (Schulman et al., 2017) to maximize both high-level and low-level objectives.

## 4 METHOD

### 4.1 MOTIVATION

While the high-level and low-level objectives in 4 and 5 focus on maximizing extrinsic team rewards, optimizing only for these objectives often leads to skill collapse, where all skills converge to similar behaviors. Each skill independently maximizes cumulative reward without explicit constraints promoting discriminability, which can result in a single skill dominating entire episodes. Under such conditions, the termination function $\beta(z, s)$ cannot effectively differentiate among skills, and the hierarchical policy loses expressive power. Prior works Eysenbach et al. (2019); Gregor et al. (2016) have addressed skill collapse by introducing intrinsic objectives that maximize the mutual information (MI) between skills and states, $I(Z; S) = H(Z) - H(Z \mid S)$, thereby encouraging skills to induce distinguishable state distributions. These approaches typically separate skill discovery and high-level optimization into two phases: skills are first discovered by mapping them to diverse states, and then the high-level policy is optimized with extrinsic reward on downstream tasks.

However, in collaborative multi-agent environments, the next state is determined jointly by the agent and the partner policy, $s_{t+1} \sim \mathcal{P}(s_{t+1} \mid s_t, a_t, a_t^p)$, so the state distribution induced by a skill $z$ depends not only on the agent policy $\pi$ but also on the partner policy $\pi^p$.

**Assumption 1:** The skill space is lower-dimensional than the joint state space, i.e., $H(Z) < H(S)$, $s \sim \rho^{\pi, \pi^p}(s)$. This reflects the fact that each skill typically corresponds to a sub-trajectory of the high-dimensional state space, allowing distinct behaviors to be captured as separate skills.

Under Assumption 1, maximizing $I(Z; S)$ is limited by the low dimensionality of the skill space. Since $H(Z)$ is fixed, the agent can achieve the maximum MI even by producing only minor, agent-centric variations that capture very little meaningful information about the partner-conditioned dynamics. For instance, consider two policies $\pi_1$ and $\pi_2$ that interact with the same partner $\pi^p$. Suppose $\pi_2$ explores the state space more broadly than $\pi_1$, which is reflected by: $H_{\rho^{\pi_1}, \pi^p}(S) < H_{\rho^{\pi_2}, \pi^p}(S)$. However, since $Z$ has fixed entropy $H(Z)$, maximizing the MI still yields

$$\max I(Z; S)_{\pi_1} = \max I(Z; S)_{\pi_2} = H(Z), \tag{6}$$

meaning that $I(Z; S)$ alone does not distinguish policies with different exploration capacities. Thus, $I(Z; S)$ provides no extra information to prefer policies that better coordinate with the partner.

Furthermore, variational approximations of $H(Z \mid S)$ typically rely on neural networks (NN) trained via cross-entropy (CE) loss, which is well known to be biased toward spurious information in feature space Wei et al. (2023). As a result, the agent can increase $I(Z; S)$ through local, repeatable

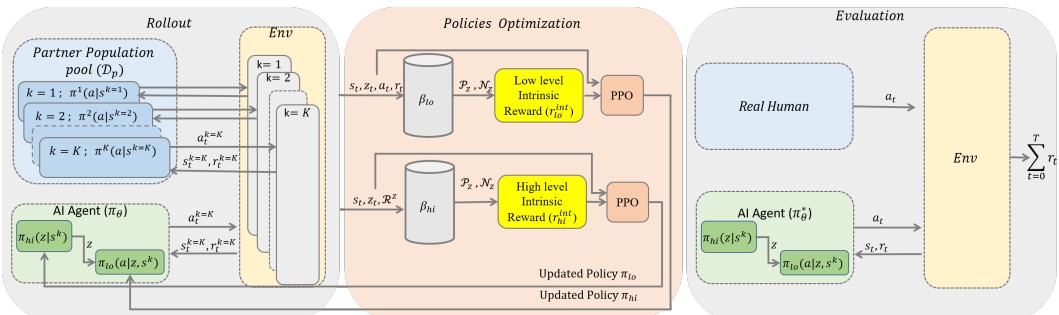

Figure 1: **Overview of PASD training and evaluation. Left:** PASD is trained with $K$ parallel rollouts, each paired with a different partner sampled from the partner pool $\mathcal{D}_p$. After every episode, high-level and low-level trajectories are stored in buffers $\beta_h$ and $\beta_l$. These trajectories are used to form *positive* pairs $\mathcal{P}_z$ (same skill across different partners) and *negative* pairs $\mathcal{N}_z$ (different skills), enabling computation of the contrastive intrinsic rewards in Eq. 13. **Right:** The optimized policies $\pi_{hi}$ and $\pi_{lo}$ are evaluated with real human partners to measure collaborative performance.

perturbations that are largely agent-centric and do not improve coordination. Formally, maximizing $I(Z;S)$ does not prevent large conditional divergences, $\mathrm{KL}\big(\pi_{hi}(\cdot \mid s_1) \,\|\, \pi_{hi}(\cdot \mid s_2)\big)$, $\quad s_1 \approx s_2$, for states corresponding to similar partner behaviour, implying that high entropy does not necessarily ensure alignment with $\pi^p$. These observations motivate a more structured objective that conditions skill discovery on partner-relevant information and encourages skills to be meaningfully distinct. In particular, skill embeddings should capture partner behaviors, ensuring that discovered skills reflect collaborative interactions rather than agent-only perturbations.

## 4.2 Partner-Adaptive Skills Discovery

In collaborative settings such as Overcooked Carroll et al. (2019), the behavior induced by a skill is shaped jointly by the agent and its partner. The same high-level skill may lead to different state transitions depending on whether the partner is fast, slow, or prioritizes different tasks. Thus, discovering meaningful skills requires capturing how a skill behaves across diverse partner policies, not just how the agent behaves in isolation. The objective in this section is to construct skill representations that remain consistent when interacting with behaviorally similar partners while remaining discriminative across different skills, ensuring that behaviorally distinct skills induce distinguishable interaction patterns. Figure 1 illustrates the overall PASD framework, showing both the training setup with partner interactions and the evaluation process with human partners. To capture partner influence as a skill discriminability measure, consider collecting $K$ parallel rollouts under the joint dynamics of the skill-conditioned agent and the partner:

$$\tau^{(k)} \sim \rho^{\pi(\cdot|s,z),\pi^P \sim D^p}(\tau), \quad k = 1, \dots, K, \tag{7}$$

where $\rho^{\pi,\pi^p}(\tau)$ denotes the trajectory distribution induced by the agent-partner interaction. Each rollout is segmented into $M$ sub-trajectories of length $L$:

$$\tau^{(k,j)} = \{s_{t_j}, a_{t_j}, \dots, s_{t_j+L}\}, \quad j = 1, \dots, M, \tag{8}$$

and representative states are sampled from each sub-trajectory, $s^{(k,j)} \sim \tau^{(k,j)}$.

**Assumption 2:** We assume that distinct sub-trajectory views of the same skill encode a consistent partner-adaptive strategy, independent of which partner $\pi^p \sim \mathcal{D}_p$ is sampled, up to stochastic noise. In other words, for any two views $(k_1, j_1) \neq (k_2, j_2)$, the additional information about the skill identity provided by one view given the other is negligible:

$$I(S^{(k_1,j_1)}; Z \mid S^{(k_2,j_2)}) \approx 0, \quad \text{or equivalently} \quad S^{(k_1,j_1)} \perp\!\!\!\perp Z \mid S^{(k_2,j_2)}. \tag{9}$$

Intuitively, once one view is observed, other views add little new information about the skill identity, reflecting reproducible partner-conditioned behavior across the partner population despite stochastic variations in trajectories. Under Assumption 2, the MI between sub-trajectory states can be used

to discover useful partner-conditioned skills i.e., skills that are both distinct with diverse partner behavior and consistent across partners showing similar behavior:

$$I(S^{(k_1,j_1)}; S^{(k_2,j_2)}) = I(S^{(k_1,j_1)}; S^{(k_2,j_2)}; Z) + I(S^{(k_1,j_1)}; S^{(k_2,j_2)} \mid Z, \pi^p), \quad (10)$$

where the first term captures *skill-discriminative information*, and the second term captures *intra-skill consistency* across similar partner behaviors.

Direct computation of the MI objective in Equation (10) is generally intractable. Following prior work van den Oord et al. (2018), we approximate it using a tractable lower bound implemented via a contrastive objective over learned state embeddings $\phi(s)$. This approximation preserves the discriminability of skills across heterogeneous partner behaviors while maintaining consistency for similar partner behaviors, and can be directly interpreted as an intrinsic reward signal to shape skill representations.

For each skill $z$, let the index set of its sub-trajectory views be $\mathcal{P}_z \equiv \{(k, j) : \tau^{(k,j)}\}$. Positive pairs are sampled from two distinct views $(k_1, j_1), (k_2, j_2) \in \mathcal{P}_z$, while negative samples are drawn from views of other skills, $\mathcal{N}_z \equiv \bigcup_{z' \neq z} \mathcal{P}_{z'}$. We approximate MI using an InfoNCE-style contrastive loss (Guo et al., 2022) over normalized embeddings $\phi(s)$, i.e., $\|\phi(s)\|_2 = 1$, so that similarities are measured on the unit hypersphere. For an anchor state $s$, with positive set $\mathcal{P}_z$ and negative set $\mathcal{N}_z$, per-anchor InfoNCE loss is:

$$\mathcal{L}_{\text{InfoNCE}} = -\frac{1}{|\mathcal{P}_z|} \sum_{s^+ \in \mathcal{P}_z} \log \frac{\exp(\text{sim}(\phi(s), \phi(s^+))/\tau)}{\sum_{s' \in \mathcal{P}_z \cup \mathcal{N}_z} \exp(\text{sim}(\phi(s), \phi(s'))/\tau)}, \quad (11)$$

where $\text{sim}(\cdot, \cdot)$ is the cosine similarity and $\tau > 0$ is a temperature parameter.

By the InfoNCE bound van den Oord et al. (2018), maximizing this reward increases a variational lower bound on the MI between the skill variable $Z$ and the state embeddings $\phi(S)$:

$$I(\phi(S); Z) \geq \log(N) - \mathcal{L}_{\text{InfoNCE}}, \quad (12)$$

where $N = |\mathcal{N}_z|$ is the number of negatives. The tightness of this approximation depends on the number of negative samples $N$ and the total number of skills $|\mathcal{Z}|$. Larger numbers of negatives and more diverse skills increase the quality of the lower bound, providing a stronger learning signal. This formulation ensures that the learned embeddings $\phi(s)$ capture both skill distinctiveness and consistency across partner behaviors. The InfoNCE objective is applied over carefully constructed positive and negative pairs. Positive pairs consist of sub-trajectories generated by the same skill interacting with different partners, which encourages embeddings to be consistent across partner behaviors. Negative pairs come from sub-trajectories of other skills, ensuring embeddings are distinct for behaviorally different skills. By maximizing InfoNCE, the learned embeddings $\phi(s)$ capture patterns that are reproducible and conditioned on the partner, rather than arbitrary agent-centric state differences. This ensures that the discovered skills reflect meaningful partner-adaptive dynamics.

## 4.3 Contrastive Intrinsic Reward

To facilitate the discovery of partner-conditioned skills, we derive an intrinsic reward by leveraging the InfoNCE objective. Specifically, the per-anchor InfoNCE probability, which measures similarity between states corresponding to the same skill relative to other skills, can be directly used as an intrinsic reward signal for both high-level and low-level policies.

For each anchor state $s \in \mathcal{P}_z$, we compute a contrastive intrinsic reward as

$$r^{\text{int}}(s) = \frac{1}{|\mathcal{P}_z|} \sum_{s^+ \in \mathcal{P}_z} \frac{\exp(\text{sim}(\phi(s), \phi(s^+))/\tau)}{\sum_{s' \in \mathcal{P}_z \cup \mathcal{N}_z} \exp(\text{sim}(\phi(s), \phi(s'))/\tau)}, \quad (13)$$

where $\phi(s)$ denotes a normalized state embedding ($\|\phi(s)\|_2 = 1$), $\text{sim}(\cdot, \cdot)$ is the cosine similarity, and $\tau > 0$ is a temperature parameter. This intrinsic reward encourages the policy to select skills that are both discriminative across heterogeneous partner behaviors and consistent across sub-trajectory views corresponding to partners with similar behaviors.

Figure 2: The five standard Overcooked layouts (left to right): Cramped Room, Asymmetric Advantages, Coordination Ring, Counter Circuit and Forced Coordination.

### 4.4 OVERALL TRAINING OBJECTIVE

To effectively learn partner-adaptive skills, we integrate the intrinsic reward defined in Equation (13) with the extrinsic environment reward in both high-level and low-level objectives mentioned in Equations (4 and 5). For the high-level manager, the intrinsic reward is accumulated and normalized over each skill segment $[t_k, t_{k+1} - 1]$:

$$\tilde{\mathcal{R}}^Z(s_{t_k}, z_k) = \frac{1}{t_{k+1} - t_k} \sum_{t=t_k}^{t_{k+1}-1} \left( (1 - \lambda)\, r(s_t, a_t, a_t^p) + \lambda\, r^{\text{int}}(s_t) \right), \tag{14}$$

where $\lambda \in [0, 1]$ controls the relative weighting of intrinsic and extrinsic rewards. The corresponding high-level objective is

$$\tilde{J}_{hi} = \mathbb{E}\left[ \sum_{k=0}^{K-1} \gamma^{t_k} \tilde{\mathcal{R}}^Z(s_{t_k}, z_k) \right]. \tag{15}$$

In the early phase of training, the intrinsic reward dominates, promoting exploration of diverse skill patterns and capturing variations in partner behaviors across different rollouts. As training progresses, the influence of the extrinsic reward gradually increases, guiding the high-level manager to refine skill selection toward maximizing task returns while maintaining consistency and discriminability across partner-conditioned interactions. For the low-level controller, the intrinsic reward is applied at each timestep:

$$\tilde{J}_{lo} = \mathbb{E}\left[ \sum_{t=t_k}^{t_{k+1}-1} \gamma^{t-t_k} \left( r(s_t, a_t, a_t^p) + \lambda\, r^{\text{int}}(s_t) \right) \right]. \tag{16}$$

Initially, the intrinsic reward drives the low-level policy to produce diverse and disentangled action distributions for each skill, capturing the variability in partner behaviors across different rollouts. As training progresses, the extrinsic reward gradually increases in influence, aligning these action distributions with task objectives while preserving the discriminability and consistency of behaviors for partners with similar tendencies. Both high-level and low-level objectives are optimized with the PPO algorithm, using the rewards defined in Equations 15 and 16. Detailed pseudocode describing the rollout procedure and the policy optimization steps of PASD is provided in Appendix A.

## 5 EXPERIMENTS

### 5.1 EXPERIMENTAL DETAILS

**Environment:** Following existing works (Strouse et al., 2021; Loo et al., 2023; Yu et al., 2023; Yang et al., 2023b), we adopt the Overcooked-AI Carroll et al. (2019) as our testbed. Overcooked-AI is a two-player cooperative benchmark derived from the Overcooked game Games (2016), in which agents collaboratively complete a soup preparation task. Agents must pick onions, place them in the pot, wait for the soup to cook, and then deliver the completed soup to the serving station, with each successful delivery yielding a reward of +20. The goal is to maximize team reward within

Table 1: Total mean reward (Mean ± Std) across three versions of each evaluation partner (early, intermediate, final checkpoint) and both starting positions.

| Method | Cramped Room | Asym. Adv. | Coord. Ring | Counter Circuit | Forced Coord. |
|---|---|---|---|---|---|
| FCP | 137.7 ± 1.0 | 90.6 ± 1.0 | 83.9 ± 5.9 | 51.3 ± 5.0 | 36.7 ± 14.4 |
| DIAYN | 33.8 ± 6.4 | 1.5 ± 0.7 | 22.5 ± 6.3 | 1.2 ± 1.0 | 1.3 ± 0.0 |
| HiPT | 117.9 ± 4.4 | 86.2 ± 0.9 | 96.0 ± 1.3 | 38.1 ± 5.3 | 35.6 ± 13.0 |
| PASD | **165.8 ± 10.0** | **145.8 ± 9.6** | **101.3 ± 8.5** | **57.37 ± 2.9** | **46.87 ± 12.3** |

a fixed episode horizon. We evaluate across five standard layouts, *Cramped Room*, *Asymmetric Advantages*, *Coordination Ring*, *Forced Coordination*, and *Counter Circuit*, illustrated in Figure 2. These layouts present diverse coordination challenges and collectively provide a widely adopted benchmark for studying partner-adaptive behaviors. Overcooked-AI is fully observable, making it a suitable testbed where coordination challenges arise solely from partner behavior rather than partial observability Yu et al. (2023); Yang et al. (2023b). For a detailed discussion of layout-specific demands, see Appendix B.

**Diverse Self-Play Partner Population:** For effective coordination with novel partners and humans, the AI agent is trained with a diverse partner population, where each partner has a unique play style and skill level. Following prior work Strouse et al. (2021); Lupu et al. (2021); Loo et al. (2023), we construct a heterogeneous policy pool. The pool consists of 16 agents trained via self-play with PPO, varying in play style and skill level. Diverse play styles are encouraged using a negative Jensen–Shannon Divergence (Loo et al., 2023), and varying skill levels are included via intermediate checkpoints (Strouse et al., 2021). During training, a partner is uniformly sampled from the heterogeneous population each episode. For evaluation with novel AI partners, a separate disjoint population of the same size is trained using the same procedure.

**Baselines** We compare PASD against standard baselines including FCP (Strouse et al., 2021), DIAYN (Eysenbach et al., 2019), and HiPT (Loo et al., 2023). Each method is trained and evaluated with the identical set of diverse partner populations introduced earlier. FCP trains the adaptive policy directly with PPO, whereas DIAYN, HiPT, and PASD adopt HRL where high-level and low-level policies are jointly optimized via the option-critic framework (Sutton et al., 1999). DIAYN uses a two-stage process where skills are first acquired through intrinsic rewards and then fine-tuned for task performance, while HiPT and PASD train both levels of the hierarchy in parallel.

**Implementation Details:** We train all methods for $10^7$ steps using 30 parallel rollouts with a horizon length of 400. For PASD, the weighting coefficient $\lambda$ is linearly annealed from 1.0 to 0.05 during training. Both low-level and high-level policies share a backbone network that consists of three convolution layers, two fully connected layers, and a recurrent LSTM layer. The network is then split into separate heads for low-level action and value prediction, and for high-level skill and value estimation. The discrete skill variable $z$ is set to dimension 6 for all layouts except *Forced Coordination*, where it is set to 5. Additional hyperparameter details are provided in Appendix D.

## 5.2 RESULTS

We organize our results into three evaluation categories based on partner type. First, the agent is paired with a diverse self-play population (Section 5.1) to test adaptation to novel AI behaviors. Second, we evaluate with human proxy models trained via behavior cloning on human–human trajectories, providing a closer approximation of human collaboration. Third, we validate performance through a controlled human-subject study to assess real human-AI coordination.

**Evaluation with Self-Play Partner Population:** We first evaluate all methods using the heterogeneous partner population, organized into three sets, early-stage, intermediate, and fully trained policies, covering a spectrum of partner proficiency from beginner to advanced. This population serves as a practical proxy for varied human collaborative behaviors (Strouse et al., 2021; Yu et al., 2023). For each set, we report the mean episodic return across all partners and starting positions,

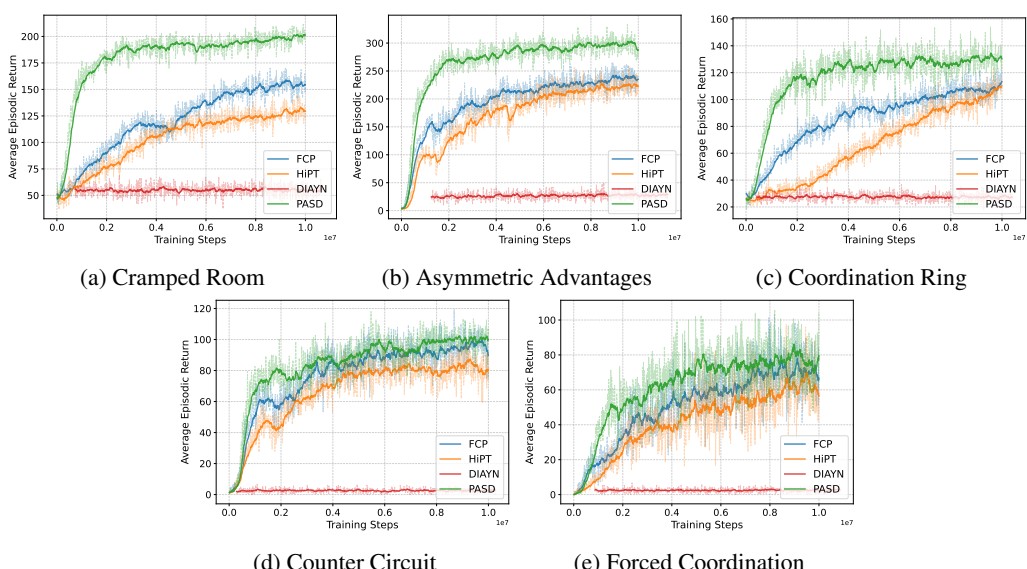

(a) Cramped Room        (b) Asymmetric Advantages        (c) Coordination Ring

(d) Counter Circuit        (e) Forced Coordination

Figure 3: Average episodic return during training across 30 parallel rollout environments.

Table 2: Total mean reward across different layouts when paired with a Behaviour Cloning (BC) partner.

| Method | Cramped Room | Asym. Adv. | Coord. Ring | Counter Circuit | Forced Coord. |
|--------|--------------|------------|-------------|-----------------|---------------|
| FCP | 118.75 | 80.00 | 79.38 | 38.13 | 30.75 |
| DIAYN | 40.00 | 0.00 | 26.25 | 1.25 | 6.30 |
| HIPT | 93.13 | 66.25 | 77.50 | 35.00 | 25.20 |
| PASD | **150.00** | **112.50** | **105.63** | **44.38** | **43.8** |

with overall performance summarized as mean ± standard deviation across the three sets (Table 1). Performance varies with layout difficulty. DIAYN performs poorly due to spurious variations disrupting skill learning. HiPT and FCP perform better but remain sensitive to redundant state information. In contrast, PASD robustly captures partner-relevant behaviors, avoids spurious variations, and achieves the highest returns across all layouts. Figures 3a–3e show average returns over 30 rollouts, illustrating that PASD converges faster and maintains stable performance across diverse partner behaviors and coordination challenges.

**Evaluation with Human Proxy Partner:** We now turn our attention to evaluating all methods with a human proxy partner trained on real human data. This proxy is obtained via behavior cloning on publicly available human–human trajectories collected by Carroll et al. (Carroll et al., 2019). Following the same procedure, the model is trained to imitate human demonstrations and used as a fixed partner during evaluation, providing a realistic approximation of human behavior under controlled conditions. Results are reported in Table 2. Performance of all methods slightly drops compared to evaluation with the self-play population, as behavior cloning with limited human data can produce policies that favor a dominant action and occasionally stall without random perturbations, as noted in (Carroll et al., 2019). Despite these challenges, PASD continues to achieve the highest returns, highlighting its ability to generalize effectively to previously unseen human-like partners.

**Human Subject Study: Real Human–AI Collaboration Evaluation** To evaluate PASD in real human–AI collaboration, we conducted a controlled human-subject study following Carroll et al. (2019). We recruited 25 participants via Amazon Mechanical Turk (AMT), each completing two episodes of 20 minutes: one paired with HiPT and one with PASD. The order of methods was randomized to mitigate ordering effects. To limit session duration and reduce participant fatigue, only HiPT and PASD were included, omitting other baselines. Participants who did not complete

Table 3: Total mean reward (Mean ± Standard Deviation) achieved by human participants when paired with HiPT and PASD across different Overcooked layouts.

| Method | Cramped Room | Asym. Adv. | Coord. Ring | Counter Circuit | Forced Coord. |
|---|---|---|---|---|---|
| HiPT | $80.00 \pm 16.34$ | $136.36 \pm 20.36$ | $46.0 \pm 12.36$ | $40.00 \pm 15.81$ | $20 \pm 0.0$ |
| PASD | $118.18 \pm 12.68$ | $198.18 \pm 19.41$ | $60.0 \pm 08.92$ | $62.5 \pm 10.95$ | $35.0 \pm 10.00$ |

both episodes were excluded, leaving 19 valid participants. Trajectories and rewards were recorded for all layouts and evaluation partners. Table 3 reports the mean ± standard deviation of total reward across conditions. Across all layouts, PASD consistently achieved higher joint rewards with human partners than HiPT, improving human–AI collaboration by 22–47%, demonstrating that partner-conditioned skill discovery meaningfully enhances real-world coordination. The full experimental setup is available at our anonymized GitHub repository [1].

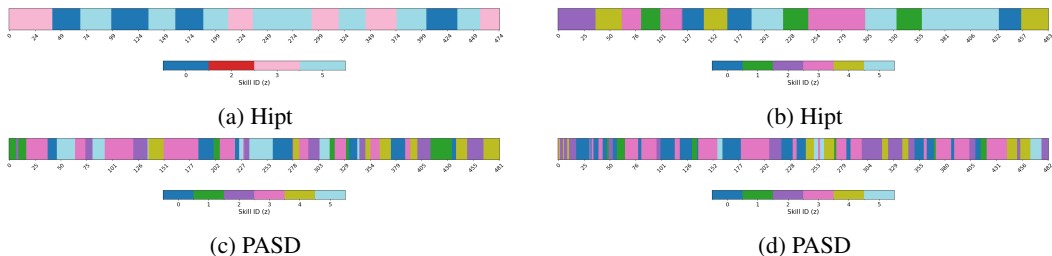

(a) Hipt                     (b) Hipt

(c) PASD                     (d) PASD

Figure 4: Skill activation over the trajectory for HiPT and PASD in Cramped Room (Left) and Coordination Ring (Right) layouts. Each colored block represents a distinct skill selected by the agent at a given timestep.

**Qualitative Analysis of Skill Disentanglement**  To illustrate how PASD (blue agent) adapts to human behaviors, we conducted controlled sessions in the Cramped Room and Coordination Ring layouts. Each session lasted 80 seconds ( 482 steps), simulating diverse gameplay for the human agent. In the Cramped Room, humans demonstrated varying preferences, e.g., picking plates from different sides or collecting soup in distinct sequences. The AI agent had to adapt by selecting the appropriate skill to minimize collisions and coordinate effectively. Similarly, in Coordination Ring, humans coordinated either clockwise or counter-clockwise, requiring agent adaptation to match the chosen pattern. Figure 4 visualizes skill usage over entire trajectories for both HiPT and PASD. PASD demonstrates distinct and stable skill activation corresponding to different human behaviors, producing sequences of atomic actions aligned with intended behavior patterns. In contrast, HiPT switches skills infrequently, often after completing entire tasks, indicating skill collapse or shortcut learning, as it fails to capture behavior-specific action patterns. Full trajectory frames and animated visualizations are available in the GitHub repository [1], illustrating how PASD adaptively selects skills to accommodate diverse human strategies throughout the episode. These results highlight PASD's ability to disentangle skills across behavioral modes, enhancing human-AI coordination. Quantitative analysis of skill variability and adaptation across partners is provided in Appendix C

## 6 CONCLUSION

This work presents PASD, a DHRL approach that introduces an intrinsic reward designed to enable effective human-AI coordination. The reward leverages a contrastive objective that encourages skill representations to be consistent across similar partners while remaining discriminative across diverse partner strategies. By capturing patterns shaped by partner behaviors, PASD promotes behavioral consistency and robustness, naturally mitigating shortcut learning that can arise from spurious information in the state space. Our experiments in Overcooked-AI demonstrate that PASD learns transferable skills that generalize across a wide range of partners, providing a foundation for more adaptive and efficient collaborative agents.

---

[1] https://anonymous.4open.science/r/pasd-22495/

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

# A  PASD – ROLLOUT AND POLICY UPDATE PROCEDURES

---

**Algorithm 1** PASD — Rollout and Intrinsic Reward Computation

---

1: **Input:** Partner population $\mathcal{D}_p$, high-level policy $\pi_{hi}(z|s)$, low-level policy $\pi_{lo}(a|s,z)$
2: **Parameters:** Intrinsic reward weight $\lambda$, rollout horizon $T$, skill segment length $L$
3: Initialize empty rollout buffers $\mathcal{B}_{hi}, \mathcal{B}_{lo}$
4: **for** each parallel rollout $k = 1, \ldots, K$ **do**
5:      Sample partner policy $\pi^p \sim \mathcal{D}_p$
6:      Reset environment $s_0 \sim \rho_0$
7:      Initialize $t \leftarrow 0$
8:      **while** $t < T$ **do**
9:          Sample skill $z_t \sim \pi_{hi}(z|s_t)$
10:          **repeat**
11:              Sample low-level action $a_t \sim \pi_{lo}(a|s_t, z_t)$
12:              Sample partner action $a_t^p \sim \pi^p(a|s_t)$
13:              Step environment: $s_{t+1}, r_t \leftarrow \text{EnvStep}(s_t, a_t, a_t^p)$
14:              Store $(s_t, z_t, a_t, r_t)$ in low-level buffer $\mathcal{B}_{lo}$
15:              Sample termination $b_t \sim \beta(z_t, s_{t+1})$
16:              $t \leftarrow t + 1$
17:          **until** termination $b_t$ or $t \geq T$
18:          Compute high-level segment reward $R^Z(s_{t_k}, z_t)$ as sum of extrinsic rewards
19:          Store $(s_{t_k}, z_t, R^Z)$ in high-level buffer $\mathcal{B}_{hi}$
20:      **end while**
21: **end for**
22: Construct positive pairs $\mathcal{P}_z$ and negative pairs $\mathcal{N}_z$ from $\mathcal{B}_{hi}$ for each skill $z$
23: Compute contrastive intrinsic rewards $r^{int}(s)$ using InfoNCE (Eq. 13)
24: Combine intrinsic and extrinsic rewards using Equations : (14, 15 and 16)
25: **Output:** High-level buffer $\mathcal{B}_{hi}$, low-level buffer $\mathcal{B}_{lo}$ with combined rewards

---

**Algorithm 2** PASD: Hierarchical Policy Update (High-level, Low-level, Termination)

---

1: **Input:** High-level buffer $\mathcal{B}_{hi}$, low-level buffer $\mathcal{B}_{lo}$, high-level policy $\pi_{hi}$, low-level policy $\pi_{lo}$, termination policy $\beta$
2: **Parameters:** PPO clipping $\epsilon$, discount $\gamma$, GAE $\lambda_{GAE}$, learning rate $\alpha$
3: **for** each gradient update iteration **do**
4:      Compute high-level advantages $\hat{A}_t^h$ from $\mathcal{B}_{hi}$ using GAE
5:      Compute low-level advantages $\hat{A}_t^l$ from $\mathcal{B}_{lo}$ using GAE
6:      Compute termination advantages $\hat{A}_t^\beta$ from $\mathcal{B}_{hi}$ or $\mathcal{B}_{lo}$
7:      Compute PPO ratio $r_t^h(\theta) = \frac{\pi_{hi,\theta}(z_t|s_t)}{\pi_{hi,\theta_{\text{old}}}(z_t|s_t)}$
8:      Compute clipped PPO loss with entropy: $L^h = -\mathbb{E}\Big[\min(r_t^h \hat{A}_t^h, \text{clip}(r_t^h, 1-\epsilon, 1+\epsilon)\hat{A}_t^h)\Big]$
9:      Compute PPO ratio $r_t^l(\theta) = \frac{\pi_{lo,\theta}(a_t|s_t,z_t)}{\pi_{lo,\theta_{\text{old}}}(a_t|s_t,z_t)}$
10:      Compute clipped PPO loss with entropy: $L^l = -\mathbb{E}\Big[\min(r_t^l \hat{A}_t^l, \text{clip}(r_t^l, 1-\epsilon, 1+\epsilon)\hat{A}_t^l)\Big]$
11:      Compute PPO ratio $r_t^\beta(\theta) = \frac{\beta_\theta(b_t|s_t,z_t)}{\beta_{\theta_{\text{old}}}(b_t|s_t,z_t)}$
12:      Compute clipped PPO loss with entropy: $L^\beta = -\mathbb{E}\Big[\min(r_t^\beta \hat{A}_t^\beta, \text{clip}(r_t^\beta, 1-\epsilon, 1+\epsilon)\hat{A}_t^\beta)\Big]$
13:      Update parameters $\theta$ of $\pi_{hi}, \pi_{lo}, \beta$ using $L^h + L^l + L^\beta$
14: **end for**
15: **Return:** Updated policies $\pi_{hi}, \pi_{lo}, \beta$

---

## B   Layout Challenges and the Need for Adaptive Skill Learning

Each Overcooked-AI layout presents unique coordination challenges requiring agents to adapt to diverse partners with varying skill levels and play styles. In *Cramped Room*, collisions are frequent due to limited space, necessitating adaptable turn-taking and collision avoidance. *Asymmetric Advantages* features partners specializing in different roles, requiring flexible skill activation for complementary behavior. *Coordination Ring* enforces a looped workflow, demanding alignment with partners' directional preferences. In *Counter Circuit*, interactions occur via counters, making timing and item exchange strategies critical. *Forced Coordination* imposes physical separation, emphasizing sequenced cooperation and dynamic routines.

In addition to the environment reward of $+20$ for each successful soup delivery, we incorporate shaped rewards to facilitate effective coordination with diverse partners. Picking up or placing an onion into a pot yields a small positive reward of $+3$, while a penalty of $-20$ is applied when the partner delivers a soup. These rewards are used in all layouts except *Forced Coordination*, where strict role asymmetry naturally enforces task specialization. By providing intermediate feedback, shaped rewards guide the agent to engage in complementary sub-tasks and adapt its behavior to align with the actions and strategies of different partners, promoting robust collaboration across all layouts.

## C   Analysis of Skill Variability and Adaptation Across Partners

We analyze the coordination behavior of PASD and HIPT by quantifying skill usage in terms of mean skill switches and mean skill entropy across partners (Figure 5). The number of skill switches captures how frequently an agent changes its skill during an episode, while the entropy measures the variability in skill selection. Higher switch counts indicate dynamic adaptation to partner actions, whereas higher entropy reflects diverse skill usage. Across these metrics, PASD consistently outperforms HIPT. In the first three layouts, skill switches are relatively infrequent, indicating less heterogeneity in the respective policy pool. The last two layouts show more frequent switches, reflecting the increased diversity in the partner policy pool, which requires PASD to adapt more dynamically. Figure 5b shows that entropy remains high across layouts, indicating that PASD effectively utilizes the full range of available skills rather than collapsing to a few, ensuring that each skill contributes meaningfully to coordination. In contrast, HIPT demonstrates lower entropy across layouts, indicating that its skill usage is more collapsed and it relies on a smaller subset of available skills rather than leveraging the full skill set.

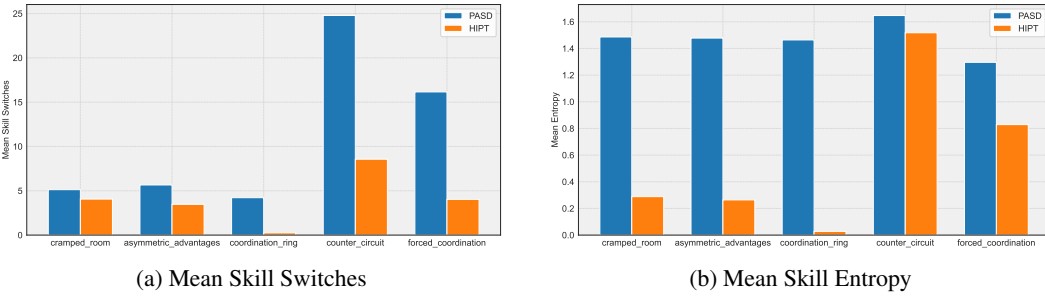

(a) Mean Skill Switches                              (b) Mean Skill Entropy

Figure 5: Mean skill switches and mean skill entropy across evaluation population pool for PASD and HIPT across different layouts.

## D   Implementation Details

We use consistent training settings across all layouts for the PPO objective. Specifically, the entropy loss coefficient is set to $0.01$ for both high- and low-level policies and linearly decays to zero over the course of training. The value function coefficient is fixed at $0.5$, and the PPO clipping coefficient is set to $0.05$. A complete summary of general hyperparameters is provided in Table 4.

Certain training parameters, such as the initial learning rate and decay schedule, are tailored to each layout to account for differing coordination challenges. Table 5 summarizes these layout-specific settings.

Table 4: Hyperparameters applied across all layouts.

| Hyperparameter | Value |
|---|---|
| Entropy coefficient | $0.01 \rightarrow 0$ (linear decay) |
| Value function coefficient | 0.5 |
| Clipping coefficient | 0.05 |
| Optimizer | Adam |
| Discount factor $\gamma$ | 0.99 |
| GAE parameter | 0.98 |
| Batch size | 64 per environment |
| Parallel environments | 30 |

Table 5: Layout-specific training parameters. The learning rate decays linearly from the initial value to the initial value divided by the decay ratio over training.

| Layout | Initial Learning Rate | Decay Ratio |
|---|---|---|
| Cramped Room | $1.0 \times 10^{-3}$ | 3 |
| Asymmetric Advantages | $1.0 \times 10^{-3}$ | 3 |
| Coordination Ring | $6.0 \times 10^{-4}$ | 1.5 |
| Forced Coordination | $8.0 \times 10^{-4}$ | 2 |
| Counter Circuit | $8.0 \times 10^{-4}$ | 3 |

