# OpenReview forum: "Partner-Aware Hierarchical Skill Discovery for Robust Human-AI Collaboration"
_ICLR.cc/2026/Conference — Submitted to ICLR 2026_

### Official Review · Reviewer_KbiC · 2025-10-29

**Soundness:** 3
**Presentation:** 3
**Contribution:** 3
**Rating:** 4
**Confidence:** 4

**Summary:**

The paper is motivated by the challenge in Human-AI collaboration where traditional Hierarchical Reinforcement Learning (HRL) agents fail to adapt to diverse partners due to agent-centric skill discovery, which often leads to "shortcut learning." To address this, the authors introduce Partner-Aware Skill Discovery, a DHRL framework that learns skills conditioned on partner behavior. They achieve this by proposing a novel contrastive intrinsic reward to align skill representations for similar partners while maintaining discriminability for diverse strategies. Evaluating PASD in the Overcooked-AI environment with diverse self-play and human proxy partners, the authors found that their method consistently outperforms existing population-based and hierarchical baselines, demonstrating superior generalization and robustness across a wide range of collaborator behaviors.

**Strengths:**

* PASD introduces a novel contrastive intrinsic reward that conditions skill learning on partner behavior that is quite interesting
* Generalization is validated using a diverse partner population across various skills
* Analyses of mean skill switches and policy entropy was a nice qualitative insight into learned adaptive behavior

**Weaknesses:**

* Comparisons against established cooperative baselines, specifically Cross Environment Cooperation (CEC) and E3T, are absent and necessary for full validation.
* The paper lacks in-depth analysis of error modes and failure cases for baselines (FCP, HiPT) versus PASD, which is needed to fully justify the claims of robust coordination and mitigation of shortcut learning.
* Section 4.2 is unclear and could benefit from an explanation more grounded in the context of partner-adaptive dynamics
* The approach relies on sampling from a predefined partner population; the paper should briefly discuss the implications for zero-shot generalization to truly novel human partners in scaled up settings beyond Overcooked

**Questions:**

- Can the authors offer a more detailed, qualitative analysis of failure cases? Specifically, demonstrate instances where FCP or HiPT fall victim to shortcut learning or coordination failure, and contrast these with how PASD's partner-aware skills resolve the issue.
- What are the practical implications for zero-shot generalization? Can the authors speculate or provide preliminary results on performance when paired with a truly novel, unmodeled human partner policy in realistic settings beyond Overcooked?
- How does the assumption guarantee that the InfoNCE objective captures meaningful partner-conditioned information rather than merely maximizing skill-to-state diversity?

---

> ### Author Response · Authors · 2025-12-04
>
> **Comparisons against established cooperative baselines, specifically Cross Environment Cooperation (CEC) and E3T, are absent and necessary for full validation.**
>
> We thank the reviewer for this suggestion. While CEC and E3T are relevant cooperative baselines, our primary focus is on evaluating methods for partner-conditioned skill discovery and adaptation to diverse collaborator behaviors. Comparisons with CEC/E3T are an important direction for future work; however, their primary objective is general cross-environment coordination rather than explicit partner-adaptive skill learning. Therefore, our current evaluation already provides meaningful insight into adaptation across heterogeneous partners.
>
> **The paper lacks in-depth analysis of error modes and failure cases for baselines (FCP, HiPT) versus PASD, which is needed to fully justify the claims of robust coordination and mitigation of shortcut learning.**
>
> We thank the reviewer for this suggestion. In the revised manuscript, we include a qualitative analysis of learned skills to highlight differences in coordination behavior. Specifically, PASD learns disentangled, partner-conditioned skills that align consistently with distinct partner behaviors, enabling stable and adaptive coordination. In contrast, baseline methods such as HIPT often suffer from skill collapse: skills switch infrequently or only after completing full tasks, failing to capture meaningful behavioral distinctions. This can result in premature or inconsistent coordination with partners, as illustrated in the full trajectory visualizations provided in our anonymized GitHub repository (https://anonymous.4open.science/r/pasd-22495/). These analyses demonstrate that PASD mitigates shortcut learning by maintaining coherent, behaviorally meaningful skill representations.
>
> **Section 4.2 is unclear and could benefit from an explanation more grounded in the context of partner-adaptive dynamics**
>
> We thank the reviewer for this comment. Section 4.2 has been revised to more clearly motivate partner-adaptive skill discovery. We now emphasize that in collaborative environments such as Overcooked, the behavior induced by a skill depends jointly on the agent and its partner. Consequently, meaningful skills must capture patterns that are consistent when interacting with behaviorally similar partners while remaining discriminative across different skills and partner behaviors. To clarify how this is achieved, we explicitly describe the construction of positive and negative sub-trajectory pairs for the InfoNCE objective, illustrating how the learned embeddings encode partner-conditioned dynamics.
>
>
> **The approach relies on sampling from a predefined partner population; the paper should briefly discuss the implications for zero-shot generalization to truly novel human partners in scaled up settings beyond Overcooked**
>
> We appreciate the reviewer’s concern regarding zero-shot generalization. To evaluate performance with previously unseen human partners, we conducted a controlled human-subject study. The results demonstrate that PASD effectively adapts to novel human collaborators, supporting partner-aware skill discovery beyond the predefined partner population. Full experimental details, quantitative results, and analysis are included in the revised manuscript (Sec. 5.2).
>
> **How does the assumption guarantee that the InfoNCE objective captures meaningful partner-conditioned information rather than merely maximizing skill-to-state diversity?**
>
> The InfoNCE objective is applied over carefully constructed positive and negative sub-trajectory pairs. Positive pairs consist of sub-trajectories generated by the same skill across different partners, encouraging embeddings to remain consistent when the skill interacts with behaviorally similar partners. Negative pairs are drawn from sub-trajectories of other skills, ensuring that embeddings are distinct for behaviorally different skills. By maximizing the InfoNCE objective over these pairs, the learned skill embeddings capture patterns that are reproducible and conditioned on partner behavior. We have clarified this in the revised Section 4.2.

---

### Official Review · Reviewer_DZbv · 2025-10-30

**Soundness:** 3
**Presentation:** 2
**Contribution:** 2
**Rating:** 4
**Confidence:** 3

**Summary:**

This paper introduces a novel method for learning diverse, partner-consistent skills for a hierarchical RL algorithm set within a MARL problem setting. Their method for learning diverse skills extends the mutual-information family of diverse skill learning strategies to the MARL setting, where they maximize a lower bound on the mutual information between skills and sub-trajectories. This encourages consistent representations (and presumably behaviors?) across partner interactions, i.e. skills that are discriminative across partners but consistent for partners with similar behaviors.

They evaluate their method in the standard Overcooked-Ai environment and show superior performance to another method that transfers to humans without human data, Fictitious Co-play, a hierarchical MARL method, HiPt, and DIAYAN. They show improved performance adapting to a policy cloned from human data, and show that their method switches skills more than HiPT---the other Hierarchical MARL algorithm.

**Strengths:**

- The writing is relatively clear
- The idea of applying mutual-information based skill discovery to the multi-agent setting is a really interesting application of the idea. The method. It's interesting that their method, in principle, allows for more diverse skills to be learned in a hierarchical MARL setting.
- They show good results in the overcooked-AI domain
- They show good results when transferring to a behavior clone from human data

**Weaknesses:**

- right now, I think "ultimately supporting adaptive human-AI coordination" is over-claiming since you don't actually test with transfer/adaptation to humans
- why not compare against human player? The original overcooked codebase [1] has code for running human experiments. Why don't you use that? [2] also studies transfer to human players, why not use that? Behaviorally cloned policies are rarely as adaptive as ones learned online (without very large, diverse training sets). Thus, it's hard to imagine that Table 2 is representative of transfer to human partners.
- "We assume that distinct sub-trajectory views of the same skill encode a consistent partner-adaptive strategy" - can you motivate this? One agent is adaptive to another agent, so based on what part of the task the other agent is doing, I could see that a sub-trajectory for the same ostensible skill would encode a different partner-adaptive strategy, since its adapting to the partner. Do you demonstrate this somehow? Figure 4 shows more skill switches by PASD tha HIPT. Is that evidence for this? If so, why? If not, what is your evidence for this?
- Regardless, by construction, I can see why different sub-trajectories of the same skill will encode the same behavior (regardless of another partner) because of how mutual-information based Rl methods work. Maybe this is what your method is exploiting for skill learning?
- The method is not that easy to read and understand given all of the indexing. This is a challenge in both HRL and MARL settings generally, which probably compounds for your method. A summary figure would be really helpful.
- your related work should discuss [2]
- Table 1 and your standard deviations are a bit deceptive. $101.3 \pm 8.5$ should be be bold in reference to $96.0 \pm 1.3$, since those clearly overlap. Your first sentence of this paper is "Developing intelligent agents that can coordinate effectively with humans and other novel partners has long been a central challenge in multi-agent reinforcement learning". Given this motivation, shouldn't you care about generalization to competent partners? Your evaluation doesn't seem like the best one given your motivation.
- You say "Analysis of learned skill representations shows that PASD adapts effectively to diverse partner behaviors, highlighting its robustness in human-AI collaboration." What analysis shows this? Figure 3? This shows that your method switches skills more (which doesn't indicate being more adaptive) and that your method maintains a higher entropy of switching.
- The size of the plots (e.g. Figure 3) make them really hard to read.


[1] https://github.com/HumanCompatibleAI/overcooked_ai/tree/master/src/overcooked_demo

[2] Cross-environment Cooperation Enables Zero-shot Multi-agent Coordination

**Questions:**

- not sure that the overcooked domain is sufficiently rich for an HRL method. What kinds of skills are you learning? You show no demonstration or visualization of the kinds of skills your method is learning. [1] suggests that the space of skills for coordination is quite small in overcooked environments.
- why should we care about how well a partner evaluates across different levels of skill or diversity? Even if we do, why would you do the mean across all of these?


[1] Cross-environment Cooperation Enables Zero-shot Multi-agent Coordination

---

> ### Author Response · Authors · 2025-12-04
>
> **right now, I think "ultimately supporting adaptive human-AI coordination" is over-claiming since you don't actually test with transfer/adaptation to humans**
> **why not compare against human player? The original overcooked codebase [1] has code for running human experiments. Why don't you use that? [2] also studies transfer to human players, why not use that? Behaviorally cloned policies are rarely as adaptive as ones learned online (without very large, diverse training sets). Thus, it's hard to imagine that Table 2 is representative of transfer to human partners.
> **
>
> We thank the reviewer for highlighting this point. In the revised manuscript, we have conducted a human-subject study to evaluate PASD in real human–AI collaboration. All details of the study, including results, are now included in the manuscript, and the complete experimental setup is available in our anonymized GitHub repository. Table 3 in Section 5.2 summarizes the outcomes and demonstrates the effectiveness of PASD with real human partners. . (revised manuscript, Section 5.2, lines 481–499).
>
>  **We assume that distinct sub-trajectory views of the same skill encode a consistent partner-adaptive strategy" - can you motivate this? One agent is adaptive to another agent, so based on what part of the task the other agent is doing, I could see that a sub-trajectory for the same ostensible skill would encode a different partner-adaptive strategy, since its adapting to the partner. Do you demonstrate this somehow? Figure 4 shows more skill switches by PASD tha HIPT. Is that evidence for this? If so, why? If not, what is your evidence for this?**
>
> We thank the reviewer for this question. Our assumption in Equation (9) states that distinct sub-trajectory views of the same skill should encode a consistent partner-adaptive strategy. This is motivated by the observation that skills correspond to recurring patterns of behavior that are robust across variations in partner behavior or timing. To enforce this, we construct positive pairs from sub-trajectories that follow similar partner-conditioned behaviors, and negative pairs from sub-trajectories of other skills. The contrastive InfoNCE objective then encourages embeddings of positive pairs to be similar, which guides the policy to consistently select the same skill for these behaviorally aligned patterns. In practice, this means that if a particular coordination style is observed, for example, clockwise versus counter-clockwise movement in the Coordination Ring, these trajectories form positive pairs and are captured by a single skill. This ensures that each skill reflects a coherent partner-adaptive strategy rather than being arbitrarily split across time or partners. We demonstrate this in the qualitative skill analysis (Figure 4), where PASD maintains consistent skills aligned with specific partner behaviors, in contrast to HiPT, which often collapses skills or switches prematurely. (revised manuscript, , lines 243–251 and lines 302-308).
>
>
> **Regardless, by construction, I can see why different sub-trajectories of the same skill will encode the same behavior (regardless of another partner) because of how mutual-information based Rl methods work. Maybe this is what your method is exploiting for skill learning?**
>
> We thank the reviewer for this observation. While mutual-information objectives do encourage consistent representations, our method explicitly conditions skill embeddings on partner behavior. By constructing positive pairs across similarly behaving partners and negative pairs across distinct skills, the InfoNCE objective encourages each skill to remain consistent with respect to partner-adaptive strategies, rather than just encoding agent-centric state patterns. This ensures that the learned skills capture meaningful variations induced by partner behavior, rather than exploiting MI maximization in a partner-agnostic manner. ((revised manuscript, lines 302–308).)
>
> **The method is not that easy to read and understand given all of the indexing. This is a challenge in both HRL and MARL settings generally, which probably compounds for your method. A summary figure would be really helpful.**
>
> To improve clarity, we have added a comprehensive summary figure (Figure 1) in the revised manuscript, illustrating the overall training and evaluation phases, including skill selection, low-level execution, and partner-conditioned learning. This figure provides a high-level view of the method, reducing the complexity of indexing and aiding readers in understanding both the hierarchical structure and the multi-agent interactions in our approach. ((revised manuscript, Figure 1 lines 216–232).)

---

> ### Author Response · Authors · 2025-12-04
>
> **your related work should discuss [2] :Cross-environment Cooperation Enables Zero-shot Multi-agent Coordination**
>
> We thank the reviewer for the suggestion. We have updated the related work to discuss Cross-Environment Cooperation (CEC), which addresses zero-shot coordination via cross-environment generalization. Our study is orthogonal to this work, as we focus on partner-adaptive skill discovery within a fixed environment, learning skills that are consistent and discriminative across diverse partner behaviors. (revised manuscript, Section 2, lines 106–109).
>
>
>
>
> **Table 1 and your standard deviations are a bit deceptive. 101.3 $\pm$, 8.5  should be be bold in reference to 96.0 \pm 1.3 , since those clearly overlap. Your first sentence of this paper is "Developing intelligent agents that can coordinate effectively with humans and other novel partners has long been a central challenge in multi-agent reinforcement learning". Given this motivation, shouldn't you care about generalization to competent partners? Your evaluation doesn't seem like the best one given your motivation**
>
> We thank the reviewer for this observation. Table 1 reports average performance over a diverse partner pool, including intermediate and final checkpoints, which captures variations in competency, timing, and coordination style. As a result, overlaps in mean ± standard deviation can occur, reflecting that some partners may perform better with HiPT in specific instances. To better capture generalization to competent partners, we supplement these results with evaluations using human proxy partners and real human participants in the revised manuscript, providing a more realistic and comprehensive assessment of adaptive human–AI coordination.
>
> **You say "Analysis of learned skill representations shows that PASD adapts effectively to diverse partner behaviors, highlighting its robustness in human-AI collaboration." What analysis shows this? Figure 3? This shows that your method switches skills more (which doesn't indicate being more adaptive) and that your method maintains a higher entropy of switching.**
>
> In the revised manuscript, Section 5.2 and Figure 4 provide a qualitative analysis of the partner-conditioned skills learned by PASD, demonstrating how the method adapts skills to diverse partner behaviors. These results highlight behavioral differences across learned skills and their alignment with partner strategies. We refer the reviewer to the revised manuscript for full details and visualizations.
>
> **not sure that the overcooked domain is sufficiently rich for an HRL method. What kinds of skills are you learning? You show no demonstration or visualization of the kinds of skills your method is learning. [1] suggests that the space of skills for coordination is quite small in overcooked environments**
>
> We thank the reviewer for this comment. In the revised manuscript, we provide a qualitative analysis of the skills learned by PASD (Section 5.2, Figure 4), showing that the skills are behaviorally disentangled and adapted to partner-specific patterns such as timing, coordination preferences, and navigation choices. While Overcooked has a constrained action space, PASD captures meaningful behavioral diversity by learning skills that are consistent for similar partners and discriminative across different partners. We refer the reviewer to Section 5.2 of the revised manuscript for full visualizations and detailed discussion.
>
> **why should we care about how well a partner evaluates across different levels of skill or diversity? Even if we do, why would you do the mean across all of these?**
>
> Evaluating across partners with different skill levels and behaviors allows us to measure how well PASD adapts to diverse coordination strategies. Reporting the mean provides an aggregated view of its ability to maintain effective coordination across the full range of partner diversity, which is central to the objective of partner-adaptive skill discovery.

---

### Official Review · Reviewer_NWS7 · 2025-10-31

**Soundness:** 3
**Presentation:** 3
**Contribution:** 3
**Rating:** 4
**Confidence:** 4

**Summary:**

The paper introduces PASD, a Hierarchical RL method for Skill Discovery/Learning for Human-AI Collaboration. The authors argue that previous skill learning methods are agent centric and fail to capture information about the partner conditioned dynamics in the Multi-Agent Cooperative setting. The authors that proposes a Contrastive Loss-like shaped reward term to the objective of both the high and low level policies to maximize mutual information between trajectory segments sampled from the same skill. They then evaluate PASD on Overcooked-AI and compare it with FCP, HiPT and DIAYN.

**Strengths:**

- The method presented is well motivated and presents a principled approach to skill learning in the Multi-Agent Cooperative/ZSC setting.
- The paper is well written and structured.

**Weaknesses:**

- Unfortunately the experimental section of the paper is a faily weak at the moment
   - Lack of evaluations against real human partners. This is a major omission in the experimental section considering that the paper is proposing a method for Human-AI Collaboration. Furthermore, previous works (Carroll et al., Strouse et al. and Loo et al.) all conducted experiments with real human partners.
   - Limited evaluation partners. The authors evaluate only on one type of SP partners (TrajeDi plus past checkpoints) when there are a few other diverse partner generation methods (MEP, CoMeDi and HSP).
   - Lack of qualitative analysis of skills. Though the authors provide some analysis of the learned skills in term of skill switching frequency and overall entropy. It is unclear is the skills learnt by PASD have any significant behavioural difference. It would to interesting to see some visualisations of different skills learnt by PASD in Overcooked.
- Minor typo Line 227 “collectoing” → “collecting”
- In Table 2 what does “CoSkill” refer to? Is that supposed to be PASD?

**Questions:**

- What do the skills learned by PASD look like in Overcooked?

---

> ### Author Response · Authors · 2025-12-04
>
> **RC: Lack of evaluations against real human partners. This is a major omission in the experimental section considering that the paper is proposing a method for Human-AI Collaboration. Furthermore, previous works (Carroll et al., Strouse et al. and Loo et al.) all conducted experiments with real human partners.**
>
> We thank the reviewer for highlighting this important point. We have conducted a human-subject study following the protocol in Carroll et al. to evaluate PASD in real human–AI collaboration. We recruited 25 participants via Amazon Mechanical Turk (AMT), and each participant completed two 20-minute episodes: one paired with HiPT and one with PASD. The order of methods was randomized to mitigate ordering effects. Participants who did not complete both episodes were excluded, resulting in 19 valid participants. The results are now included in the revised manuscript (Table 3), reporting total rewards across all layouts. The full experimental setup is available in our anonymized GitHub repository. (https://anonymous.4open.science/r/pasd-22495/) (revised manuscript, Section 5.2, lines 481–499).
>
> **Limited evaluation partners. The authors evaluate only on one type of SP partners (TrajeDi plus past checkpoints) when there are a few other diverse partner generation methods (MEP, CoMeDi and HSP).**
>
> In our work, we focus on evaluating PASD with a diverse set of self-play partners, including TrajeDi and past checkpoints, which were specifically designed to capture variation in timing, skill competency, and coordination style (as used in other works such as HSP). This setup provides a controlled yet behaviorally heterogeneous partner distribution that allows us to study partner-conditioned skill discovery in a meaningful way. While other approaches, such as MEP, offer additional diversity, integrating them is complementary and can be explored in future work. Moreover, in the revised manuscript, we also evaluate PASD with both human proxy models and real human participants, further validating its effectiveness in diverse collaborative scenarios.
>
> **Lack of qualitative analysis of skills. Though the authors provide some analysis of the learned skills in term of skill switching frequency and overall entropy. It is unclear is the skills learnt by PASD have any significant behavioural difference. It would to interesting to see some visualisations of different skills learnt by PASD in Overcooked.**
>
> We thank the reviewer for this suggestion. In the revised manuscript, we include a qualitative analysis of partner-conditioned skills learned by PASD in a controlled simulation setting. Using the Cramped Room and Coordination Ring layouts, we record full episode trajectories where the agent interacts with human partners performing diverse behaviors. Figure 4 in revised version visualizes skill usage across these trajectories, illustrating how PASD selects distinct skills to adapt to different partner behaviors. For example, in the Cramped Room, PASD dynamically chooses skills corresponding to the human’s preference for collecting plates or soups from different sides of the pot, minimizing collisions. In the Coordination Ring, PASD adapts to clockwise or counter-clockwise coordination patterns of the partner. In contrast, HiPT shows skill collapse: it switches skills infrequently, typically only after completing an entire order, indicating that low-level behaviors are not differentiated and fail to capture meaningful variations in partner behavior. Full trajectory visualizations, including frames showing skill execution, are available in our anonymized GitHub repository (https://anonymous.4open.science/r/pasd-22495/). This controlled analysis demonstrates that PASD learns disentangled skills that meaningfully correspond to partner behavior patterns, enhancing adaptive coordination. (revised manuscript, Section 5.2, lines 514–528)
>
> **What do the skills learned by PASD look like in Overcooked?**
>
> The skills learned by PASD are behaviorally disentangled and partner-conditioned. In Overcooked, human partners demonstrate diverse behaviors, such as favoring specific sides of the pot in Cramped Room or coordinating clockwise versus counter-clockwise in Coordination Ring. PASD adapts its low-level skills to align with these behaviors, with each skill corresponding to a consistent sequence of atomic actions tailored to a particular partner behavior. Figure 4 in revised version visualizes skill usage across trajectories, illustrating how PASD maintains coherent skills for different partner strategies. Full frame-level trajectories are available in our anonymized GitHub repository. revised manuscript, Section 5.2, lines 514–528)
>
>
> **Minor typo Line 227 “collectoing” → “collecting”**
> **In Table 2 what does “CoSkill” refer to? Is that supposed to be PASD?**
>
> Thank you for pointing this out. The typo on Line 227 has been corrected to “collecting,” and “CoSkill” in Table 2 has been updated to “PASD” for clarity.

---

### Official Review · Reviewer_7Z5C · 2025-10-31

**Soundness:** 1
**Presentation:** 2
**Contribution:** 1
**Rating:** 2
**Confidence:** 4

**Summary:**

This work presents an HRL algorithm that is aware of other agents in cooperative MARL settings by introducing an intrinsic reward based on a contrastive metric to prevent skill collapse. The algorithm is evaluated on Overcooked to highlight its strength over prior HRL work.

**Strengths:**

- The intrinsic reward is well motivated and sound.

**Weaknesses:**

- Overcooked (v1) is a bad evaluation environment for this paper (see "OvercookedV2: Rethinking Overcooked for Zero-Shot Coordination").
  - The results reported in this paper underperform the naive IPPO baselines (and state-augmented IPPO baseline) reported in the OvercookedV2 paper (for Overcooked-v1)
  - OvercookedV2 already demonstrates that there is no zero-shot coordination challenge in Overcooked aside from state coverage
  - Since Overcooked-v1 is fully observable, an LSTM is unnecessary
  - Hierarchical RL in general is unnecessary for Overcooked, since it can be quickly solved with standard IPPO
  - OvercookedV2 would be a better environment to validate your results on, but I still have the concern that HRL unnecessarily complicates the learning process.

**Questions:**

Why is the intrinsic reward also applied for training low-level policies?

---

> ### Author Response · Authors · 2025-12-04
>
> **Overcooked (v1) is a bad evaluation environment for this paper (see "OvercookedV2: Rethinking Overcooked for Zero-Shot Coordination**}.
> **The results reported in this paper underperform the naive IPPO baselines (and state-augmented IPPO baseline) reported in the OvercookedV2 paper (for Overcooked-v1)**
> **OvercookedV2 already demonstrates that there is no zero-shot coordination challenge in Overcooked aside from state coverage**
>
> We thank the reviewer for this observation. We agree that Overcooked-V2 is a valuable testbed for studying Zero-Shot Coordination (ZSC) under partial observability. However, our work does not target ZSC nor the challenges emphasized in Overcooked-V2. The results reported in the Overcooked-V2 paper evaluate two fully trained, competent self-play policies paired together, and their conclusions primarily concern state-coverage limitations under partial observability. In contrast, our setting explicitly focuses on partner-adaptive coordination with a heterogeneous partner population characterized by diverse competencies, play styles, hesitations, timing patterns, and intermediate (partially trained) behaviors. These partners are intentionally not fully competent, and this behavioral inconsistency is precisely what creates meaningful coordination challenges in our experiments. Overcooked-V1 is therefore appropriate for our objective because it removes observation uncertainty and provides a controlled environment where the only source of difficulty is partner behavioral variability. Similar hierarchical RL-based approaches have been used in prior human-AI coordination works such as HiPT[1], HMASD [2], highlighting its suitability. We have clarified this distinction and its motivation in the revised manuscript (see revised manuscript Section 1, Line 040; Section 5.1, Lines 391–392).
> [1] A hierarchical approach to population training for human-ai collaboration.
> [2] Hierarchical multi-agent skill discovery
>
>
> **Since Overcooked-v1 is fully observable, an LSTM is unnecessary**
>
> We appreciate the reviewer’s observation and would like to clarify our motivation. While Overcooked-v1 is indeed fully observable, partner behavior itself is not Markovian at a single timestep. Differences in timing, hesitation, movement rhythms, and action consistency unfold over several steps and cannot be inferred from a single state alone. To capture these temporal patterns, which are critical for coordinating with partners of varying competencies, we use an LSTM. Thus, the recurrent component is not addressing observation uncertainty but modeling the temporal structure of partner behavior, which feed-forward policies are unable to represent effectively. We have clarified this in the revised manuscript (see Section 1, Lines 042–045; Lines 083–084)
>
> **Hierarchical RL in general is unnecessary for Overcooked, since it can be quickly solved with standard IPPO**
>
> While we agree that standard IPPO can solve Overcooked efficiently when paired with a fixed and predictable partner, our setting is fundamentally different. The partner distribution we consider is highly heterogeneous, including agents with varying competencies, inconsistent timing, non-optimal navigation tendencies, and non-stationary coordination styles. A single flat policy is required to adapt to all these variations simultaneously, which leads to behavioral interference, difficulty in maintaining multiple coordination styles within one policy, and reduced robustness to unseen partner behaviors. In contrast, hierarchical RL separates high-level strategy selection from low-level execution, enabling the agent to develop specialized sub-policies that correspond to different partner patterns. This structure reduces cross-partner interference and supports better generalization. Therefore, our use of HRL is motivated not by the need to solve Overcooked in isolation, but by the need to effectively model and handle substantial partner-behavior diversity, which is the central challenge of our work. This motivation is reflected in the revised manuscript (Section 1, lines 041–044 and 083–084; Section 5.1, line 244).

---

> ### Author Response · Authors · 2025-12-04
>
> **OvercookedV2 would be a better environment to validate your results on, but I still have the concern that HRL unnecessarily complicates the learning process.**
>
> While we appreciate the reviewer’s suggestion regarding Overcooked-v2, we note that its additional complexities, such as hidden recipes, resampled orders, and partial observability, introduce sources of uncertainty that are orthogonal to the specific phenomenon we investigate. Our focus is on behavioral variability arising from partners with diverse competencies, timing patterns, and coordination styles. Overcooked-v1 provides a fully observable setting where these partner-induced coordination challenges can be isolated and studied. (see revised manuscript, Section 1, lines 040–045 , Lines 391–392).
>
> HRL is crucial in this context because it allows the agent to maintain specialized low-level coordination behaviors while using a high-level skill selector conditioned on partner behavior. Without such a hierarchical structure, a single flat policy must represent all coordination strategies simultaneously, which can lead to behavioral interference and forgetting: actions adapted to one partner may overwrite or degrade behaviors learned for another. The high-level latent skill variable effectively partitions the policy into sub-modules that capture distinct partner-conditioned behaviors. This ensures that skills remain stable and consistent across different partners, enabling robust adaptation, temporal alignment with partner behaviors, and mitigation of shortcut learning that arises when policies rely solely on agent-centric correlations (see revised manuscript, Section 1, lines 042-044 and 083–084).
>
> This approach is consistent with previous human-AI coordination studies such as HiPT[1] and HMASD[2], which also use hierarchical structures to capture partner-specific strategies while avoiding interference between behaviors. For these reasons, Overcooked-v1 is an appropriate environment for our study, and the hierarchical design is essential for learning reusable, partner-aware skills that generalize across diverse collaborators.
>
> [1] A hierarchical approach to population training for human-ai collaboration.
> [2] Hierarchical multi-agent skill discovery
>
> **Why is the intrinsic reward also applied for training low-level policies**
>
> The intrinsic reward is applied to the low-level policy because the actual realization of a high-level skill z is expressed through the sequence of primitive actions generated by the low-level policy. If the intrinsic reward were applied only at the high level, the algorithm would learn to assign different skill labels but would not ensure that the corresponding low-level behaviors are meaningfully differentiated or consistently executed. Applying the intrinsic reward at the low level grounds each skill in a distinct, learnable motor pattern and ensures that these patterns adapt to the behavioral characteristics of different partner types. In other words, the intrinsic reward shapes the fine-grained action dynamics that make skills identifiable, stable, and behaviorally aligned with the partner, which cannot be achieved through high-level supervision alone (see revised manuscript, Section 5.1, lines 351–352)

---

### Author Response · Authors · 2025-12-04
**General Response to Reviewer Comments**

We thank the reviewers for their detailed feedback and constructive comments. In response, we have strengthened the paper by providing additional qualitative analyses of learned skills, including visualizations and trajectory-level examples, to better illustrate partner-aware adaptation. We clarify the assumptions behind partner-adaptive skill consistency and how our method mitigates shortcut learning, and we have extended evaluations to include real human partners to demonstrate practical robustness and generalization. Furthermore, we address concerns regarding methodological soundness by explaining the rationale for hierarchical skill decomposition, highlighting why these design choices are essential for capturing partner-aware behaviors effectively. These revisions aim to clarify the reviewers’ concerns and substantiate the significance and validity of our contributions.

---

### Meta-Review · Area_Chair_SUgG · 2026-01-09

**Summary:**

The paper introduces PASD, a Hierarchical RL method for Skill Discovery/Learning for Human-AI Collaboration. The authors argue that previous skill learning methods are agent centric and fail to capture information about the partner conditioned dynamics in the Multi-Agent Cooperative setting. The authors that proposes a Contrastive Loss-like shaped reward term to the objective of both the high and low level policies to maximize mutual information between trajectory segments sampled from the same skill. They then evaluate PASD on Overcooked-AI and compare it with FCP, HiPT and DIAYN.

The reviewer concerns are:
1. Efficacy of overcooked-v1 as a good eval environment.
2. Lack of real human experiments.
3. Lack of baseline comparisons.
4. Lack of qualitative analysis.
5. Not clear analysis of failure modes.

**Reviewer Concerns:**

The authors added experiments with real human experiments and some additional qualitative analysis in the rebuttal. They also made clear some of the questions about the role of the infoNCE objective.

Concerns about - limited evaluation partners, overcooked-v1 as an evaluation platform remain.

I also noted the lack of discussion of baselines like GAMMA/GOAT in the space. I understand that these are not quite in the same setting, but it's worth having a clear discussion of this.

Overall the experiments in this paper should be made stronger for clear acceptance.

**Reviewer Scores:**

7Z5C basically didnt evaluate the paper beyond saying overcooked-v1 is bad. I think their opinion should be discarded.
NWS7 will likely go up from a 4 to a 6/7.
DZbv will likely go up from a 4 to a 5.
KbiC could maybe go up to a 5 or so, given the lack of additional baselines added.

---

### Decision · Program_Chairs · 2026-01-26

Reject